# Reinforcement Learning Enables Real-Time Planning and Control of Agile Maneuvers for Soft Robot Arms

**Rianna Jitosho**[*],  **Tyler Ga Wei Lum**[*],  **Allison M. Okamura**,  **C. Karen Liu**

Stanford University

{rjitosho, tylerlum, aokamura, karenliu}@stanford.edu

**Abstract:** Control policies for soft robot arms typically assume quasi-static motion or require a hand-designed motion plan. To achieve real-time planning and control for tasks requiring highly dynamic maneuvers, we apply deep reinforcement learning to train a policy entirely in simulation, and we identify strategies and insights that bridge the gap between simulation and reality. In particular, we strengthen the policy's tolerance for inaccuracies with domain randomization and implement crucial simulator modifications that improve actuation and sensor modeling, enabling zero-shot sim-to-real transfer without requiring high-fidelity soft robot dynamics. We demonstrate the effectiveness of this approach with experiments on physical hardware and show that our soft robot can reach target positions that require dynamic swinging motions. This is the first work to achieve such agile maneuvers on a physical soft robot, advancing the field of soft robot arm planning and control. Our code and videos are publicly available at https://sites.google.com/view/rl-soft-robot.

**Keywords:** Soft Robotics, Reinforcement Learning, Sim-to-Real Transfer, Dynamics and Controls

## 1  Introduction

Compared to traditional rigid robot manipulators, soft robot arms with inherent material compliance offer improved adaptability and safety in tasks requiring contact with the environment [1, 2, 3]. However, soft robots have yet to achieve the high-speed capabilities of rigid robots, which can leverage dynamics to achieve difficult tasks [4, 5, 6, 7, 8]. We aim to enable dynamic control for a class of soft robot arms made from an inflated fabric tube with a fabric Pneumatic Artificial Muscle (fPAM), which has near-linear dynamics and high-bandwidth dynamic response with minimal hysteresis [9].

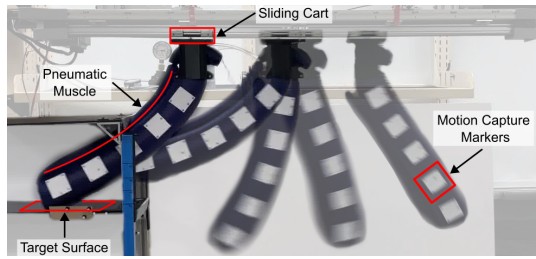

Figure 1: We achieve soft robot arm control tasks that require leveraging robot dynamics. Our inflated-beam robot uses a pneumatic bending actuator and a 1D mobile base to perform agile maneuvers such as swinging onto a target surface. We use a motion capture system to measure the robot's state and compute control actions with a policy trained using reinforcement learning.

We mount this inexpensive and lightweight inflated-beam soft robot arm on a mobile platform that allows 1D linear actuation of the arm's base (Fig. 1). Integrating the soft arm with a sliding base increases the robot's workspace and allows additional dynamics that can be leveraged by a controller.

Most prior work in soft robot arm control assumes quasi-static motion [10, 11, 12, 13]. This assumption is enticing because soft robots have complex dynamics resulting from their compliance and high dimensionality, so using quasi-static models for soft robot motion makes them easier to

---

[*] Equal Contribution.

7th Conference on Robot Learning (CoRL 2023), Atlanta, USA.

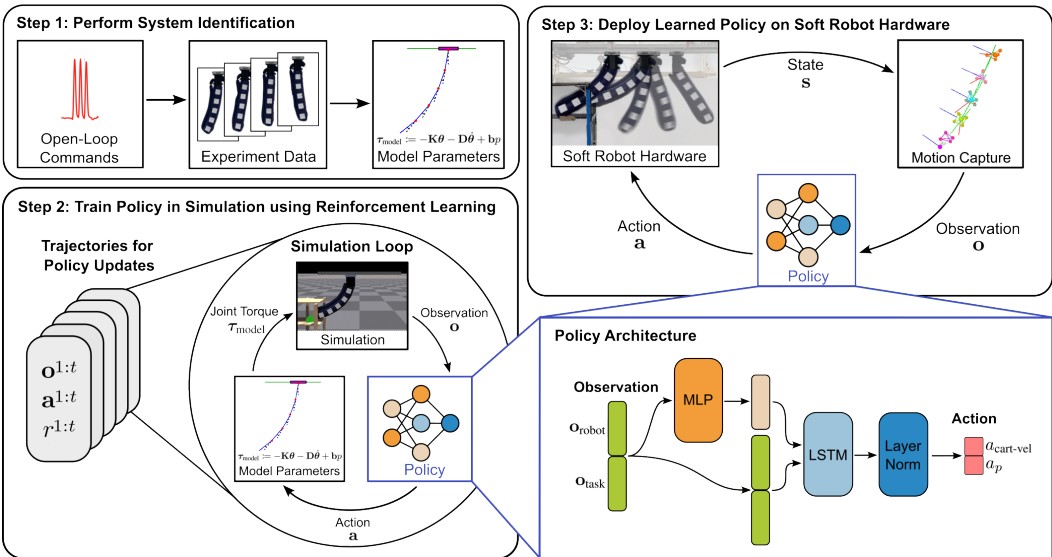

Figure 2: In our proposed framework, we first fit parameters for a physics-based dynamic model using about 30 seconds of real experimental data. Next, we train a control policy using deep reinforcement learning entirely in simulation using the fit model from the previous step. Finally, we deploy our learned control policy directly on real hardware using a motion capture system to measure the robot's state.

control. However, this approach is unable to exploit dynamic capabilities, such as leveraging swinging motions to generate momentum. More recent works have incorporated dynamic models [14, 15], but limit their applications to tasks that are simple enough to use hand-designed motion plans, such as tip trajectory following. In contrast, our learned policies eliminate the need for a pre-defined motion plan because they are goal-conditioned and handle planning and control jointly.

In our work, we use a dynamic model that approximates our soft robot arm as a multi-link rigid body system. This allows us to leverage robot dynamics, as well as existing rigid-body simulators and control frameworks, to achieve tasks requiring high-speed motion. With this model, we employ reinforcement learning (RL) to train a control policy and deploy this policy directly onto a real soft robot system. However, naive transfer to the real world is unsuccessful, which is a common challenge encountered when deploying control policies learned in simulation [16, 17]. We overcome this challenge by implementing key adjustments to the simulator's actuator and sensor models as well as improving the policy's robustness to modeling inaccuracies using domain randomization. Fig. 2 illustrates the three-step process of our proposed framework. To our knowledge, this is the first work that achieves real-time motion planning and control for high-speed behaviors on a physical soft robot arm. This work demonstrates that a simple physics-based model can be used to build a simulator that enables learning of agile maneuvers for soft robots with zero-shot sim-to-real transfer.

To evaluate our system's performance, we optimize trajectories for each goal in our test set as an approximate upper bound of performance. We find that the policy from our RL framework performs similarly to the approximate upper bound while achieving real-time planning and control. In contrast, trajectory optimization requires an offline planning stage that takes several minutes to perform, which is impractical for real-world use.

## 2 Related Work

Here we discuss the most relevant prior work (see Supplementary Material (SM) A for more detail).

**Soft Robot Arm Modeling.** Our work requires motion that is impossible under quasi-static assumptions, preventing the use of kinematic models such as the piecewise constant curvature model [13]. However, we find that we do not require the higher model fidelity offered by the augmented rigid body formulation [18], Cosserat rod theory [19], Koopman [20] models, or Long-Short Term Memory models [14]. Instead, we choose a model similar to the n-link pendulum model used by Jitosho et al. [21] since it is cheap to compute, simple to implement, and enables fast simulation.

**Soft Robot Arm Control.** Recent works have demonstrated dynamic behaviors on soft robot arms. Centurelli et al. [14] train a control policy using reinforcement learning to perform tip path following under light payloads. Fischer et al. [15] use an operational space controller and proportional-derivative control to perform tip path following for pick and place, throwing, and drawing. Bruder et al. [20], Haggerty et al. [22] use Koopman-based system identification to construct explicit linear dynamical models of soft robots, and they use model-predictive control and a linear quadratic regulator, respectively, to perform trajectory following tasks. Dynamic models improve the speed and accuracy of tip path following, but these works require hand-designed tip trajectories, limiting the capabilities of their robots. In our work, we achieve autonomous motion planning. This enables agile maneuvers to emerge during training, such as high-speed swinging motions, that would be difficult to design by hand.

**Reinforcement Learning for Controlling Physical Robots.** A common training paradigm when learning policies for physical robots is to train control policies entirely in simulation and then deploy them in the real world [23]. However, policies trained in simulation often fail in the real world due to model discrepancies between the simulated and real systems [16, 17]. One approach to overcoming this sim-to-real gap is training the policy under domain randomization so that the resulting policy will be robust to simulation inaccuracies [24]. This approach has been successfully applied to robotics domains such as legged locomotion [25, 26, 27, 28, 29], in-hand manipulation [30, 31, 32], and robot grasping [24, 33, 34]. In addition to this randomization, Tan et al. [35] overcome the sim-to-real gap by improving their simulator with system identification and actuator modeling for their quadruped robot. Our approach to actuator modeling achieves sim-to-real transfer despite lacking access to the underlying dynamics of our actuators.

# 3 Method

## 3.1 Soft Robot Modeling

We consider a class of inflated-beam soft robots that utilize fPAMs as their bending actuators [9]. Our hardware implementation features one beam as the main body and a second inflated beam with a smaller radius (the fPAM) attached along the length of the main body. Pressurizing the fPAM causes one side of the main body to shorten, forming a bend (Fig. 3a,b).

We model the soft robot arm as an n-link pendulum with hinge joints between links and between the base link and the sliding cart (Fig. 3c). This allows us to model the soft robot arm with a finite number of state variables and use rigid-body simulators. The cart is attached to fixed rails with a prismatic joint. We model each hinge joint as a rotational spring and damper to capture the stiffness and damping of the inflated beam. We assume a linear mapping between the pressure inside the bending actuator and the

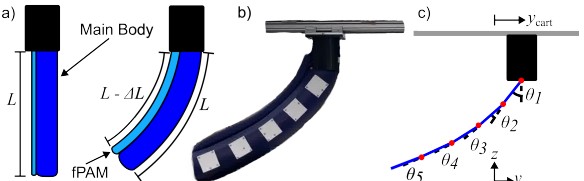

Figure 3: a) Illustration of the fabric Pneumatic Artificial Muscle (fPAM) actuation. Inflating the fPAM causes it to shorten by $\Delta L$. This enforces an asymmetric shortening of the robot's main body, causing it to bend. b) Image of our physical soft robot arm mounted to the mobile cart of a linear actuator. c) Illustration of the 5-link pendulum model. Rigid links (blue) are connected via hinge joints (red) to each other and to the cart (black). This cart is attached to the rails of the linear actuator (gray) with a prismatic joint. We define the cart position $y_{\text{cart}}$ and relative joint angles $\theta_i$. We also model rotational springs and dampers at each hinge joint (not visualized).

torque applied at each hinge joint. This results in the following definition for the hinge joint torques:

$$\boldsymbol{\tau}_{\text{model}} := -\mathbf{K}\boldsymbol{\theta} - \mathbf{D}\dot{\boldsymbol{\theta}} + \mathbf{b}p, \tag{1}$$

where $\boldsymbol{\tau}_{\text{model}} \in \mathbb{R}^n$ are the joint torques, $\boldsymbol{\theta} \in \mathbb{R}^n$ are the joint angles, $\dot{\boldsymbol{\theta}} \in \mathbb{R}^n$ are the joint velocities, $p \in \mathbb{R}$ is the fPAM pressure, $\mathbf{K} \in \mathbb{R}^{n \times n}$ is a diagonal matrix for stiffness, $\mathbf{D} \in \mathbb{R}^{n \times n}$ is a diagonal matrix for damping, and $\mathbf{b} \in \mathbb{R}^n$ is a linear mapping from bending actuator pressure to joint torques. We set $n = 5$ and fit $\mathbf{K}$, $\mathbf{D}$, and $\mathbf{b}$ using data from experiments with open-loop commands. We use measurements from the physical hardware for mass and inertia values. We describe these measurements as well as our system identification method, results, and validation in SM-B.

## 3.2 Learning Dynamic Soft Robot Control Policies

In this work, we focus on two soft robot arm control tasks:

1. *Free Space Target Reaching:* Reaching tip position targets in free space. We are particularly interested in targets that are high enough that the soft arm requires swinging to achieve the task.
2. *Shelf Target Reaching:* Swinging onto the surface of a shelf (Fig. 1). This task is difficult because the robot needs to coordinate the swinging of the arm with the sliding of the base such that 1) the tip swings above the shelf surface while avoiding a collision, and 2) the tip moves laterally over the shelf edge before the tip swings back down. This task also demonstrates that we can readily apply our framework to tasks involving contact.

We formulate these as RL problems, in which an agent sequentially chooses actions to interact with an environment with the goal of maximizing cumulative reward [36]. For each task, we learn a single policy that can reach a diverse range of target positions, enabling real-time planning. We describe our RL framework here, and provide problem formulation and policy training details in SM-C.

**Action Space.** The RL agent's action $\mathbf{a} \in \mathbb{R}^2$ includes the commands for our robot's two actuators. The first actuator is a pressure regulator that controls the fPAM pressure. We define the commanded fPAM pressure in kPag as $a_p \in \mathbb{R}$, which is different from the fPAM pressure $p$ in Eq. 1 because it takes time for the regulator to inflate or deflate the fPAM. The second actuator is the linear actuator controlling cart motion. The linear actuator has an internal controller which operates in "Velocity Mode", which requires the user to send velocity commands over time. We define the commanded cart velocity in m/s as $a_{\text{cart-vel}} \in \mathbb{R}$, which is different from the actual cart velocity $v_{\text{cart}} \in \mathbb{R}$ because it takes time to accelerate the cart to a new velocity. Section 3.3 provides a comprehensive description of the actuator dynamics in our simulator, emphasizing the crucial additional details concerning the real actuator dynamics that needed to be accurately modeled in the simulation.

**Observation Space.** The RL agent's observation $\mathbf{o}$ consists of two parts: $\mathbf{o}_{\text{robot}}$ and $\mathbf{o}_{\text{task}}$. $\mathbf{o}_{\text{robot}} \in \mathbb{R}^8$ includes the cart position $y_{\text{cart}} \in \mathbb{R}$, the cart velocity $v_{\text{cart}} \in \mathbb{R}$, the tip position $\mathbf{x}_{\text{tip}} \in \mathbb{R}^2$, the tip velocity $\mathbf{v}_{\text{tip}} \in \mathbb{R}^2$, the previous cart velocity command $a_{\text{prev-cart-vel}} \in \mathbb{R}$, and the current fPAM pressure $p \in \mathbb{R}$. We found that including $a_{\text{prev-cart-vel}}$ and $p$ improves system performance, and we hypothesize these observations encode critical information for understanding actuator dynamics and generating coordinated actions over time. $\mathbf{o}_{\text{task}}$ is defined specifically for each task. For the free space target reaching task, we define $\mathbf{o}_{\text{task}} \coloneqq \mathbf{x}_{\text{target}}$, where $\mathbf{x}_{\text{target}} \in \mathbb{R}^2$ is the target position. For the shelf target reaching task, we define $\mathbf{o}_{\text{task}} \coloneqq (\mathbf{x}_{\text{target}}, d_{\text{target}})$, where $\mathbf{x}_{\text{target}} \in \mathbb{R}^2$ is the target position and $d_{\text{target}} \in \mathbb{R}$ is the difference between the y-coordinates of the target and edge of the shelf (Fig 4).

**Reward Function and Reset Conditions** We describe the reward function and reset conditions in SM D.

**Architecture** Our control policy consists of a multi-layer perceptron (MLP) followed by a Long Short-Term Memory (LSTM) layer. The MLP has 3 layers (256, 128, 64 hidden units) connected with Exponential Linear Unit (ELU) layers. The output of this MLP is concatenated with the original input, which is then fed into an LSTM layer with 256 hidden units followed by a Layer Normalization layer. This recurrent architecture allows the policy to leverage temporal information to improve performance and handle the partial observability of the robot state.

Figure 4: Shelf target reaching task in simulation. $\mathbf{x}_{\text{target}}$ is the target position, which is visualized as a green sphere with radius $d_{\text{success}} = 0.04$ m. $d_{\text{target}}$ is the difference between the y-coordinates of $\mathbf{x}_{\text{target}}$ and the edge of the shelf.

## 3.3 Sim-to-Real Policy Transfer

One of the main contributions of this work is overcoming the discrepancy between the robot's simulated training environment and the real robot hardware. We find that matching the actuation and sensing in simulation to the behaviors of their real-world counterparts is crucial for achieving zero-shot sim-to-real transfer, while a high-fidelity dynamic model of the soft robot is not necessary. In addition, we concur with previous work that

modeling latency in actuators as well as domain randomization also play important roles in the success of sim-to-real transfer [35].

**Pressure Regulator Dynamics.** Physical pressure regulators have finite flow rates, resulting in reduced tracking performance for high-frequency signals. As an example, the frequency response of the pressure regulator developed by Gao et al. [37] shows decreasing gain with increasing frequency past a threshold. This issue is also apparent in our hardware, where the pressure regulator struggles to track abrupt pressure command changes (ex. black and red signals in Fig. 5). As in [37], we find our pressure regulator to behave like a low-pass filter, so we model this behavior in simulation by applying an infinite impulse response (IIR) filter to the pressure command. We implement this using the update rule:

$$p^{k+1} \leftarrow \alpha a_p^k + (1 - \alpha)p^k,$$

where $\alpha \in \mathbb{R}$ is an empirically-chosen parameter. This model closely approximates reality (Fig. 5) and serves as a general approach for modeling pressure regulators in soft robot RL frameworks.

**Linear Actuator Dynamics.** Many actuators come with internal controllers that make them easier to use (e.g. position control for motors), but introduce modeling challenges for sim-to-real transfer. For example, our linear actuator's "Velocity Mode" controller hides the actual cart physics from the user. This controller first requires the user to specify a fixed acceleration $\dot{v}_{\text{fixed}} \in \mathbb{R}$ before running the control policy. Then, the user sends velocity commands over time, and the actuator accelerates the cart at the fixed acceleration until the velocity reaches the latest commanded velocity. This results in a piece-wise linear velocity profile (e.g. red signal in Fig. 6) consisting of constant slopes (constant acceleration) of either 0 or $\pm\dot{v}_{\text{fixed}}$ (e.g. we set $\dot{v}_{\text{fixed}} = 6$ m/s$^2$).

In simulation, we can directly manipulate the cart's kinematic state. However, the real cart cannot instantaneously reach any velocity due to physical limitations. Additionally, directly imposing a new cart state could result in significant simulation inaccuracy because the soft robot arm is attached to the cart (could lead to very unrealistic forces on the soft robot arm). To bridge simulation and reality, we design a simulated cart controller that operates by applying forces in a manner that closely approximates the behavior of the real cart controller. Importantly, this approach does not require a precise understanding of the physical linear actuator controller's implementation. Because there are forces on the cart from the rest of the robot, we cannot simply apply a constant force to the cart to achieve constant acceleration. We address this challenge by implementing a low-level cart controller in simulation that performs acceleration tracking when the velocity error is high and velocity tracking when the velocity error is low. We implement this by applying a cart force (in Newtons) $F_{\text{cart}} \in \mathbb{R}$ defined as:

$$F_{\text{cart}} = \begin{cases} m\dot{v}_{\text{fixed}} + k_{\dot{v}}(\dot{v}_{\text{fixed}} - \dot{v}_{\text{cart}}), & |e_v| > 0.1 \text{ (track acc)} \\ k_v e_v, & |e_v| \leq 0.1 \text{ (track vel)} \end{cases}$$

$$e_v = a_{\text{cart-vel}} - v_{\text{cart}} \quad \text{(Note: } a \text{ is for action, not acceleration)}$$

where $e_v \in \mathbb{R}$ is the velocity tracking error, $m \in \mathbb{R}$ is the mass of the robot, $\dot{v}_{\text{fixed}} \in \mathbb{R}$ is the fixed cart acceleration, $\dot{v}_{\text{cart}} \in \mathbb{R}$ is the actual cart acceleration, and $k_{\dot{v}} \in \mathbb{R}$ and $k_v \in \mathbb{R}$ are the

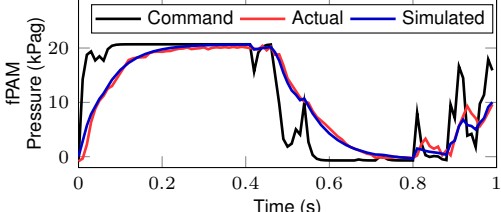

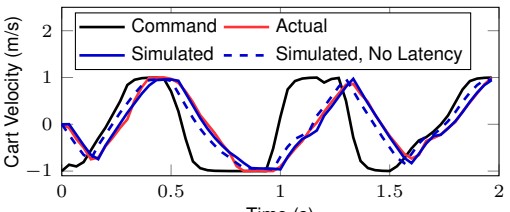

Figure 5: Pressure in the fPAM in response to a pressure command. The fPAM pressure regulator is bandwidth-limited, so we simulate this behavior by applying an infinite impulse response (IIR) filter to the RL command. The RMS error between the actual and simulated pressure is 0.94 kPag, which is small compared to the range of pressure commands (-0.7 to 20 kPag).

Figure 6: Cart velocity in response to a velocity command. The linear actuator cannot respond immediately to changes in commanded velocity, so we simulate the linear actuator as an acceleration tracker for large velocity errors and a velocity tracker for small errors. The RMS error between the actual and simulated velocity is 0.08 m/s, which is small compared to the range of velocity commands (-1.0 to 1.0 m/s). The simulated velocity trajectory more closely approximates the real velocity trajectory when actuator latency is simulated.

acceleration tracking gain and velocity tracking gain, respectively. In the acceleration tracking case, $m\dot{v}_{\text{fixed}}$ is a feedforward term based on $F = ma$, and $k_v(\dot{v}_{\text{fixed}} - \dot{v}_{\text{cart}})$ is a feedback term to improve tracking on this fixed acceleration. In the velocity tracking case, the velocity is already close to the commanded velocity, so only a small feedback term is required. Our simulated cart motion closely approximates the real cart's piece-wise linear velocity (Fig. 6). This strategy can be adapted to overcome sim-to-real difficulties for a broad range of actuators.

**Linear Actuator Latency.** Physical actuators exhibit latency, which can hinder sim-to-real transfer when left unmodeled in simulation [35]. We model the linear actuator latency in our simulator by adding a time delay between the time the policy outputs a velocity command and the time that the command is sent to the simulator's lower-level controller. Fig. 6 shows that modeling the delay helps our simulated cart's velocity profile closely match the real cart's velocity profile.

**Velocity Measurements.** During physical experiments, we compute tip velocity with first-order finite-differencing, so we replicate this in our simulated observations.

**Domain Randomization.** We use domain randomization while training our policy in simulation so that it can handle the remaining sim-to-real gap not captured by our other adjustments. We apply additive Gaussian noise of $N(0, \sigma_{\text{obs}})$ and $N(0, \sigma_{\text{act}})$ for all elements of the observation $\mathbf{o}$ and the action $\mathbf{a}$, respectively, where $\sigma_{\text{obs}} \in \mathbb{R}$ and $\sigma_{\text{act}} \in \mathbb{R}$ are the observation and action noise parameters. We also apply uniform scaling of $U(1 - \epsilon_{\text{dyn}}, 1 + \epsilon_{\text{dyn}})$ to all elements of the dynamics parameters $\mathbf{K}, \mathbf{D}, \mathbf{b}$, where $\epsilon_{\text{dyn}} \in \mathbb{R}$ is the dynamics noise parameter. We tune these parameters so that the policy takes longer to train, but the policy is still able to achieve nearly the same reward when fully trained (additional domain randomization details in SM F).

## 4 Results

We evaluate the capabilities of our RL framework by performing demonstrations and experiments on real soft robot arm hardware (hardware details in SM-E). Our learned policies achieve real-time planning and control of agile maneuvers not previously demonstrated on soft robot arms.

### 4.1 Characterizing Performance for Free Space Target Reaching

In this section, we assess the quality of behaviors generated by our RL framework by showing that its performance is close to that of an approximate upper bound method. The upper bound method uses trajectory optimization (TO) for motion planning and a real-time tracking controller for control (details in SM-H). We emphasize that trajectory optimization is not a practical choice for our task since it requires a time-consuming offline planning stage.

We compare our RL method with the TO method on the free space target reaching task. When training our RL method, we sample target positions $\mathbf{x}_{\text{target-train}} \in [-0.4\text{ m}, 0.0\text{ m}] \times [0.55\text{ m}, 0.7\text{ m}]$. We test our RL method on an array of 54 target positions $\mathbf{x}_{\text{target-test}} \in [-0.4\text{ m}, 0.0\text{ m}] \times [0.55\text{ m}, 0.8\text{ m}]$ (Fig. 7). Targets with z-coordinates above 0.7 m are outside the training distribution. To compare with the approximate upper bound, TO, we precompute 54 trajectories in an offline planning stage, each specifically optimized to reach its respective target position ($\sim$4.7 hours to precompute, as solving each trajectory took between 1-13 minutes, depending on the target position). All targets are on the left side of the soft robot because the fPAM is designed to bend towards this side.

Fig. 7 shows that our learned policy reaches a wide range of target positions, missing only the most difficult ones. Specifically, targets towards the upper left are difficult because they require the most aggressive swinging to reach, and targets with y = 0 are difficult because the robot needs to swing aggressively to the sides in order to swing the tip towards the rail center (y = 0) while avoiding the rail limits. Despite this, our policy achieved a success rate of 94% for targets within the training distribution, 50% for targets outside the training distribution, and 80% overall. The TO method achieved success rates of 94%, 83%, and 91%, respectively. This suggests that our RL method matches the performance of the TO method within its training distribution, and also has some level of generalization to targets outside of this range, though at a much lower success rate.

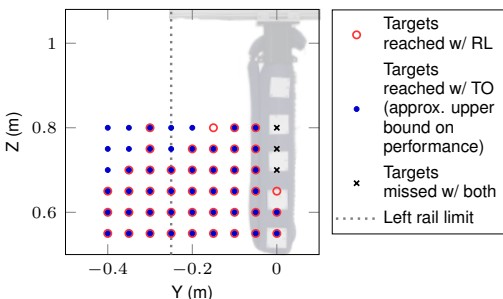

Figure 7: Free space target reaching on hardware for our learned policy (RL) vs. trajectory optimization (TO). Targets above 0.7 m are outside our learned policy's training distribution. We overlay the starting position of the soft robot arm used for each trial.

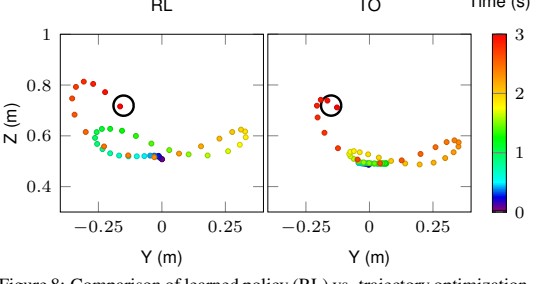

Figure 8: Comparison of learned policy (RL) vs. trajectory optimization (TO) for reaching a target tip position in free space. The black circle is centered at the target position with a radius of 4 cm. Both methods perform high-speed swinging to reach the target position, showing that RL is able to produce motions that are comparable to TO.

Fig. 8 compares tip trajectories achieved on hardware with our RL method and the TO method. These plots show similar trajectories for both methods, suggesting that our RL method approaches the approximate upper bound of performance. We see that the robot reaches the target tip position by building up momentum over multiple swings to increase its tip height. This demonstrates that the free space target reaching task is not possible without well-timed actuation that leverages swinging motion. Such a motion plan would be difficult to design by hand, but our RL method enables real-time planning and control of these types of agile maneuvers for soft robot arms.

We further demonstrate that leveraging swinging is required for task success by comparing our policy to a PID controller, which does not reason about swinging (method and results in SM-I).

## 4.2 Ablation Study on Shelf Target Reaching

We assess the significance of individual components of our framework through an ablation study with the shelf-reaching task. During training, we sample target positions $\mathbf{x}_{\text{target-train}} \in [-0.48 \text{ m}, -0.4 \text{ m}] \times [0.58 \text{ m}, 0.67 \text{ m}]$ and position the shelf such that $d_{\text{target}} \in [-0.05 \text{ m}, 0.2 \text{ m}]$ (Fig 4). To evaluate each ablation, we deploy the policy on physical hardware and run 3 trials for each of the following shelf target positions (y, z): $(-0.48 \text{ m}, 0.6 \text{ m})$, $(-0.48 \text{ m}, 0.63 \text{ m})$, $(0.48 \text{ m}, 0.66 \text{ m})$, and $(-0.53 \text{ m}, 0.63 \text{ m})$. We then compute the success rate across these 12 trials and compare the results of these policies with our best-performing policy (Table 1).

**Actuator and Sensor Models.** We find that our actuator and sensor modeling improvements in our simulator (Sec. 3.3) were crucial for success. We verify this with the following ablations:

- *No fPAM Smoothing:* We remove the IIR filter that models the fPAM pressure regulator's dynamics, so pressure commands instantly update the pressure.
- *No Cart Acceleration Tracking:* We remove the acceleration tracking mode in our low-level controller, leaving only the velocity tracking mode.
- *No Cart Action Delay:* We remove the delay in sending the RL command to the simulated low-level cart controller.
- *Ground Truth Velocities:* We use ground truth velocities in our observations instead of velocities computed using first-order finite differencing.

Table 1: Shelf Target Reaching Task Ablation Study

| Policy | Success Rate |
|---|---|
| Ours | 92% |
| *Actuator and Sensor Model Ablations* | |
| No fPAM Smoothing | 0% |
| No Cart Acceleration Tracking | 8% |
| No Cart Action Delay | 0% |
| Ground Truth Velocities | 0% |
| *Domain Randomization Ablations* | |
| Domain Randomization on Observations | 25% |
| Domain Randomization on Dynamics | 25% |
| No Domain Randomization | 17% |
| *Architecture Ablations* | |
| MLP Only | 0% |
| LSTM Before MLP | 0% |
| *fPAM Actuation Ablation* | |
| Cart Actions Only (No fPAM) | 0% |

While these policies obtain high success rates in simulation, these ablations result in poor performance on the physical robot due to the larger sim-to-real gap. The failures appear similar to those we saw during initial development before we added these sim-to-real adjustments. We found the underlying causes of these failures by methodically evaluating the failures and comparing the sim and real observations, actions, and dynamics. More details about this process can be found in SM G.

**Domain Randomization.** We find that the best performance on hardware is achieved by training a policy with domain randomization on policy actions. We consider ablations with different domain randomization schemes to verify this:

- *Domain Randomization on Observations:* We remove randomization on the actions and add Gaussian noise to the observations.
- *Domain Randomization on Dynamics:* We remove randomization on the actions and scale dynamic parameters $(\mathbf{K}, \mathbf{D}, \mathbf{b})$ by a value from a uniform distribution.
- *No Domain Randomization:* We remove randomization on the actions.

Within this ablation subset, removing all domain randomization performed the worst, verifying the benefit of domain randomization in crossing the sim-to-real gap. The ablations with observation and dynamics randomization each show a small improvement in success rate compared to the ablation without any randomization, suggesting there is a small sim-to-real gap in our dynamic parameters and our observation measurements. However, there is a large difference in success rate between our best-performing policy and these ablations, which suggests that action randomization is important to address the remaining sim-to-real gap not captured by our adjustments described in Sec. 3.3.

**Architecture.** Our best-working policy architecture is a multi-layer perceptron (MLP) followed by a Long-Short Term Memory (LSTM) network. We consider ablations with different architectures:

- *MLP only:* We remove the LSTM layer so that the policy does not have memory.
- *LSTM before MLP:* We modify the architecture so that the LSTM is run before the MLP.

Both ablations perform poorly on physical hardware. We expected the MLP without the LSTM to perform worse because memory has been shown to be useful in control tasks with partial observability [38, 39]. The higher success rate for the architecture with the LSTM after the MLP suggests that the MLP provides a useful representation for the LSTM to model temporal dependencies.

**fPAM Actuation.** We verify that the fPAM is necessary for task success by running an ablation in which we turn off fPAM actuation in both the simulated and real systems (i.e. the robot is limited to cart actions only). The robot had low success rates in both simulation and on physical hardware, demonstrating that both actuators (fPAM and cart) are required for the shelf-reaching task.

## 5   Limitations

While our work demonstrates significant progress in enabling real-time planning and control for soft robot arms, there are limitations that should be addressed in future work. One limitation is the use of motion capture, which is costly to set up and not suitable for outdoor or field use. Onboard state estimation would allow broader use of our system. Another limitation is that we do not fine-tune the policy in the real world, which could allow adaptation to changes in the environment or robot hardware. Finally, while our work demonstrates that using simple physics-based models for this class of soft robots is an effective approach, it is important to acknowledge that the implementation in this work has fewer actuators than many soft robot systems. More complex soft robots may benefit from higher-fidelity or learning-based models. In future work, we are interested in exploring how to expand this framework to other soft robot platforms.

## 6   Conclusion

In this study, we demonstrate that deep reinforcement learning enables real-time planning and control of dynamic maneuvers for soft robot arms. We identify key insights needed to overcome sim-to-real challenges for zero-shot sim-to-real transfer, including domain randomization and improvements to actuator and sensor modeling. Our work provides evidence that accurate actuator and sensor models are crucial for task success, and that agile behaviors can be learned without requiring a high-fidelity model of soft robot dynamics. We demonstrate the effectiveness of our framework through physical experiments that show our soft robot arm executing agile maneuvers in real time.

**Acknowledgments**

If a paper is accepted, the final camera-ready version will (and probably should) include acknowledgments. All acknowledgments go at the end of the paper, including thanks to reviewers who gave useful comments, to colleagues who contributed to the ideas, and to funding agencies and corporate sponsors that provided financial support.

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

# Supplementary Material

## A Additional Related Work

**Soft Robot Arm Modeling.** While the pose of any point on a traditional, rigid-link robot can be fully defined by its link lengths and joint angles, the kinematics of soft robots are more complex due to their elasticity and continuum nature [1]. While Finite Element Methods (FEM) provide high-accuracy models of soft robots, their computational cost and high dimensionality make them difficult to use in control methods [2]. Instead, researchers opt for more tractable approximations for practical implementation [1]. The most common modeling approximation is the Piecewise Constant Curvature (PCC) model, a kinematic model in which the robot is approximated as a series of constant-curvature arcs. Other works use models that offer a middle ground in computational complexity. Katzschmann et al. [3] introduce an augmented rigid body formulation that models a soft robot as a rigid-bodied robot with parallel elastic actuation, which enables a dynamic model that respects PCC assumptions. Naughton et al. [2] draws from Cosserat rod theory to model soft arms as continuum elements that can bend, twist, shear, and stretch.

Another approach to dynamics modeling is through data-driven modeling methods. Bruder et al. [4] model a soft robot using the Koopman Operator, in which a projected linear state space model is fit with collected data. Jitosho et al. [5] create a soft robot simulator that models soft growing robots as $N$-link pendulums with linear springs and dampers between links, and prismatic joints to model lengthening and retracting, and these parameters are fit with collected data. Bern et al. [6] train a neural network to find a mapping from motor angles to quasi-static tip position. Centurelli et al. [7] model soft robot forward dynamics with a Long Short-Term Memory (LSTM) network to find a mapping from an actuation vector to a tip position.

**Soft Robot Sim-to-Real** Addressing the sim-to-real gap is an important challenge in the domain of soft robots. This challenge arises from the complex kinematics and dynamics inherent to these systems, rendering their accurate simulation notably more difficult compared to conventional rigid robots. Many works use FEM to accurately simulate the behavior of real soft robot hardware. Dubied et al. [8] model a soft robot arm with pneumatic actuation using differentiable FEM, which provides gradients to efficiently calibrate the Young's modulus, fit damping, and tune actuation parameters to match the real system. Hagiwara et al. [9] develop a soft robotic gripper with both soft deformation and high stiffness, and they model it with finite element analysis to support simple grasping with visual servo control. Zhang et al. [10] use a differentiable simulator to learn the material parameters of a soft robotic fish from quasi-static data and use these parameters to create robotic fish tail designs that can be used in the real world. These soft robot modeling methods can be very accurate and useful for sim-to-real transfer, but are too slow to be used for training a policy using reinforcement learning.

Sim-to-real transfer of control policies for soft robots is a promising direction of research. Li et al. [11] use a deep deterministic policy gradient (DDPG)-based control system for a soft robot arm to perform path following, which is first trained in a kinematics-only simulator and then fine-tuned on real-world data. Although they successfully transfer policies to the real world, they only perform slow-moving path following, and their kinematics-only simulator has a large sim-to-real gap, so they need multiple rounds of real-world fine-tuning (deploy policy, collect data, retrain, repeat) to overcome the sim-to-real gap. Similarly, Zhang et al. [12] use Q-Learning to train a control policy for a slow-moving reaching task, and their policy needs to be trained both in simulation and in the real world. In contrast, our control policies demonstrate highly dynamic swinging behavior, are trained entirely in simulation, and can be deployed on a real soft robot zero-shot.

There are many other aspects of soft robot sim-to-real that are being studied. Yoo et al. [13] propose a neural network to predict a soft robot's current shape configuration from point cloud data, and they demonstrate strong performance in the real world when trained entirely on simulated data. An exciting direction for future work is to replace the motion capture system used in our work

with this neural network, as it would allow our control policies to be deployed in environments that do not support motion capture hardware. Kriegman et al. [14] introduce a soft robot design and construction kit that can be used to simulate, create, and measure the simulation-reality gap of simple soft robots. While this approach can identify soft robot designs with smaller sim-to-real gaps, our work is focused on overcoming the sim-to-real gap when deploying learning-based control policies on real soft robot hardware.

**Soft Robot Arm Control.** Many previous works leverage traditional, physics-based methods to control soft robot arms. Santina [15] achieves closed-loop stabilization with feedback linearization on a simulated soft inverted pendulum. Weerakoon and Chopra [16] use an energy-based controller for a soft robot swing-up task and then use a linear-quadratic regulator controller for stabilizing the soft robot in the upright position. Bruder et al. [4] use model predictive control with a Koopman model to perform tip trajectory tracking for a soft robot. Grube et al. [17] perform soft robot arm trajectory following with a kinematic controller and a dynamic controller, and they find that the dynamic controller achieves higher accuracy and robustness than the kinematic controller but also requires more computational resources. Haggerty et al. [22] model and control soft robots using a data-driven learning method based on Koopman operator theory, allowing them to generate fast motion with high-deflection shapes.

Another method for soft robot arm control is training a neural network with large amounts of data. Bern et al. [6] train a neural network to approximate the forward dynamics of a soft robot arm, and then perform quasi-static trajectory following by using gradient-based optimization with this learned model. Thuruthel et al. [19] train a recurrent neural network to model the forward dynamics of a soft robot and then use the learned model and trajectory optimization to create open-loop trajectories. Next, they test these open-loop trajectories on the real robot and use this data to train a neural network in a supervised fashion to be used as a closed-loop predictive controller. Qiuxuan et al. [20] fit a soft robot dynamics model with a multi-layer perceptron and then train a control policy with deep Q-learning.

We would like to note the distinction between soft robot arms and rigid robot arms with soft actuation. The actuation choice (soft vs. traditional) has minimal impact on the robot kinematics, whereas the limb choice (soft vs. rigid) greatly affects robot kinematics due to the compliance of the soft body. This results in additional modeling and control challenges for soft robot arms.

**Reinforcement Learning for Controlling Physical Robots.** An alternative paradigm for training robot control policies using reinforcement learning involves learning from both simulation and real-world data. Bousmalis et al. [21] first train a control policy in simulation and then fine-tune the policy on a real robot. Rusu et al. [22] use real-world data to train a generator network that transforms simulated images into real images so that the policy can learn from more realistic observations. While this paradigm can reduce the sim-to-real gap, we do not use this approach for our problem because collecting large amounts of real-world data would be time-intensive and result in significant degradation to the hardware.

There is a growing interest in applying reinforcement learning to soft robot arm control, with existing works primarily focusing on trajectory following at relatively slow speeds [2, 7, 19, 23, 24]. Naughton et al. [2] focus on additional tasks that require maneuvering between structured obstacles. Similarly, we achieve tasks that require reasoning about objects in the robot's environment. Bianchi et al. [25] develop an open-loop controller for throwing objects to target positions by learning a model between the sequence of actuations and the resulting landing position. Our work differs from previous implementations of RL for soft robot arms because we focus on achieving tasks that require high-speed motion and do not need a predefined motion plan.

## B   System Identification Method, Results, and Verification

We use measurements from the physical hardware to compute mass and inertia values for our robot model. Our physical soft robot arm contains internal hardware inside the tip, but otherwise is a

hollow, inflated beam. Based on this, we approximate the most distal link as a solid cylinder since it contains internal hardware, and we approximate the proximal links as cylindrical shells. We measure the mass of the full robot arm and of the internal hardware, and use this to compute the weight of the distal cylinder and each of the four cylindrical shells (0.1 kg and 0.05 kg respectively). With the measured masses and the measured cylinder radius (0.038 m), we can compute the inertia for each geometry accordingly.

We fit $\mathbf{K}$, $\mathbf{D}$, and $\mathbf{b}$ using data from a fixed-base experiment (no use of the cart's linear actuator). In this experiment, we send a sequence of fPAM pressure commands and measure the resulting sequence of pressure values ($p^{1:N}$) as well as the resulting robot trajectory ($\boldsymbol{\theta}^{1:N}$). From these values, we can compute velocities ($\dot{\boldsymbol{\theta}}^{1:N}$) and accelerations ($\ddot{\boldsymbol{\theta}}^{1:N}$) via 4th-order finite differencing.

Using these values, we compute the joint torques $\boldsymbol{\tau}^k$ at timestep $k$ via inverse dynamics [26]:

$$\boldsymbol{\tau}^k = \mathbf{M}(\boldsymbol{\theta}^k)\ddot{\boldsymbol{\theta}}^k + \mathbf{C}(\boldsymbol{\theta}^k, \dot{\boldsymbol{\theta}}^k)\dot{\boldsymbol{\theta}}^k + \mathbf{g}(\boldsymbol{\theta}^k), \tag{2}$$

where $\mathbf{M}$, $\mathbf{C}$, and $\mathbf{g}$, are functions that compute the mass matrix, Coriolis terms, and torques due to gravity, respectively.

To fit model parameters $\mathbf{K}$, $\mathbf{D}$, and $\mathbf{b}$, we pre-compute ($\boldsymbol{\tau}^{1:N}$) using Eq. 2, then we fit model parameters using least squares on our definition of joint torques in Eq. 1:

$$\min_{\mathbf{K},\mathbf{D},\mathbf{b}} \sum_{k=1}^{N} \| -\mathbf{K}\boldsymbol{\theta}^k - \mathbf{D}\dot{\boldsymbol{\theta}}^k + \mathbf{b}p^k - \boldsymbol{\tau}^k \|^2. \tag{3}$$

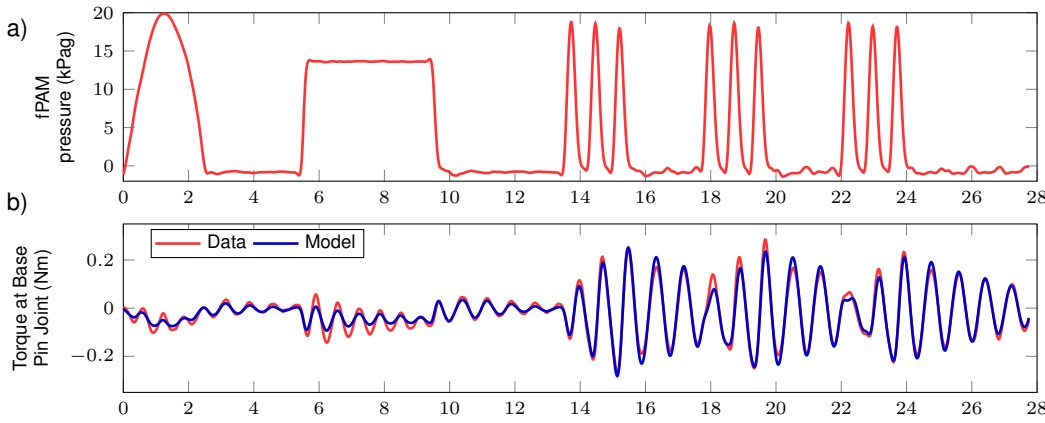

Figure 9: Results of the system identification fitting procedure. a) Input control signal into the system b) Comparison of joint torques computed from experiment data (red) versus our model with fitted parameters (blue). The RMS error is 0.02 Nm, which is small compared to the torque values in the plot.

Fig. 9 shows the results of the system identification procedure. Fig. 9a shows the sequence of fPAM commands sent to the physical robot. It includes a slow sweep of pressures to capture quasi-static behavior, a square wave to capture the step-response, and a series of high-frequency sine waves to capture swinging dynamics. The fPAM commands were sent at 60 Hz for roughly 28 s, resulting in about 1600 timesteps of measurements. Fig. 9b shows the results of the least squares fitting. We overlay the joint torques derived from measured data (Eq. 2) versus the one computed using estimated parameters (Eq. 1). The RMS error is 0.02 Nm, which is small compared to the actual torque values (-0.28 to 0.25 Nm), showing good agreement between our model and the measurements.

The numerical values for stiffness, damping, and our control mapping are below:

$$diag(\mathbf{K}) = (0.8385, 1.5400, 1.5109, 1.2887, 0.4347) \text{ Nm/rad}$$
$$diag(\mathbf{C}) = (0.0178, 0.0304, 0.0528, 0.0367, 0.0223) \text{ Nm*s/rad}$$
$$\mathbf{b} = (0.0247, 0.0616, 0.0779, 0.0498, 0.0268) \text{ Nm/psi}$$

To demonstrate that our dynamic model captures our soft robot arm's behaviors, we compare the tip trajectories of the real robot with a simulation that uses our fit model. The fPAM command sequence sent to each system is a sinusoid with a frequency not used in the system identification procedure. Fig. 10 shows the comparison of tip position over time. The tip y-coordinate ranges from -10 to 7 cm with an RMS error of 2 cm. The tip z-coordinate ranges from 51 to 52 cm with an RMS error of 0.3 cm. The average distance between the simulated and real robot tip positions across all time steps is 2 cm. We find that this model fidelity is sufficient for the tasks demonstrated in our work.

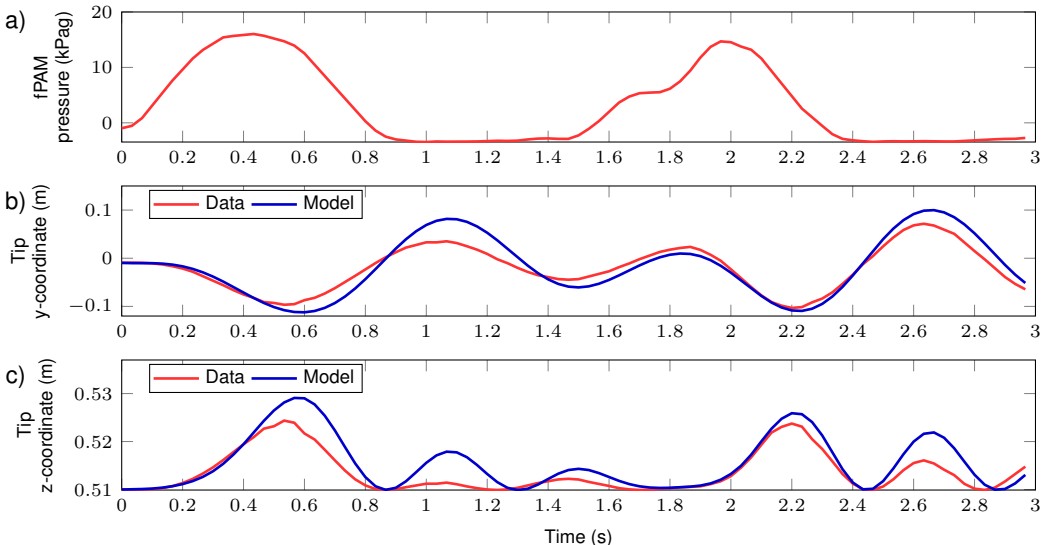

Figure 10: Evaluation of fit model. a) Input control signal into the system that is different than the original system identification experiment. b) Comparison of tip y-coordinate between experiment data versus a simulation with the fit model. The RMS error is 2 cm. c) Comparison of tip z-coordinate between experiment data versus a simulation with the fit model. The RMS error is 0.3 cm.

## C   RL Framework Details

**Problem Definition for Reinforcement Learning.** We formulate the soft robot arm control task as a reinforcement learning problem. This is commonly modeled as a Markov Decision Process (MDP) given by $(\mathcal{S}, \mathcal{A}, \mathcal{P}, \mathcal{R})$, where $\mathcal{S}$ is the state space, $\mathcal{A}$ is the action space, $\mathcal{P} : \mathcal{S} \times \mathcal{A} \times \mathcal{S} \to \mathbb{R}$ is the transition function, and $\mathcal{R} : \mathcal{S} \times \mathcal{A} \times \mathcal{S} \to \mathbb{R}$ is the reward function. $\mathcal{P}(s_{t+1}|s_t, a_t)$ gives the probability of the agent transitioning from state $s_t$ to $s_{t+1}$ when it takes the action $a_t$. $\mathcal{R}(s_t, a_t, s_{t+1})$ gives the reward $r_t$ the agent receives when it transitions from state $s_t$ to $s_{t+1}$ when it takes the action $a_t$. The agent's goal is to maximize the return $R_t = \sum_{k=t}^{\infty} \gamma^{k-t} r_k$, which is the total discounted reward from timestep $t$ onwards, where $\gamma \in [0, 1]$ is the discount factor that defines how much the agent favors near-term rewards over far-term rewards.

This formulation assumes a fully observable MDP, but in many real-world robotics control problems, the full state of the robot cannot be captured. These problems can be modeled as a Partially-Observable Markov Decision Process (POMDP), where the agent receives observations from an observation model $\mathcal{O}(o_t|s_t, a_t)$. In this setting, the agent cannot observe the full state at each timestep. Common solutions to this problem include stacking a history of observations [27] or compressing the history into a hidden state through the use of recurrent neural networks [28, 29]. These solutions have the added benefit of access to additional temporal information.

## D   Reward Function and Reset Conditions

We define our reward at each timestep, $r \in \mathbb{R}$, to be a weighted sum of rewards from individual reward functions $r = \sum_i w_i r_i$, where $r_i \in \mathbb{R}$ is the reward from the $i$-th reward function and

$w_i \in \mathbb{R}$ is weight associated with the $i$-th reward function. Specifically, we use:

$$w_{\text{pos}} = 1.0, \quad r_{\text{pos}} = \begin{cases} 1000, & ||\mathbf{x}_{\text{tip}} - \mathbf{x}_{\text{target}}|| < d_{\text{success}} \\ 0, & \text{otherwise} \end{cases}$$

$$w_{\text{vel}} = 0.1, \quad r_{\text{vel}} = ||\mathbf{v}_{\text{tip}}||$$

$$w_{\text{limit}} = 1.0, \quad r_{\text{limit}} = \begin{cases} -100, & |y_{\text{cart}}| > \bar{y} \\ 0, & \text{otherwise} \end{cases}$$

where $d_{\text{success}} = 0.04$ m is the maximum tip-to-target distance to count as a success and $\bar{y} = 0.25$ m is the rail limit. The small velocity reward improves training speed because it encourages swinging, which is required for higher target positions. An episode terminates when the cart exceeds the rail limits, the tip position reaches the target position, or the episode exceeds 100 timesteps (3 seconds).

## E    Experimental Setup

To demonstrate our RL framework on physical hardware, we design and build an experimental setup featuring a soft, inflated-beam robot with a mobile base (Fig. 1). The central component is a computer running the control policy. It sends pressure commands to the soft robot, sends velocity commands to the linear actuator, and receives state measurements from a motion capture system. Despite not having a GPU, policy inference takes under 10 ms running on this computer, which enables real-time planning and control.

**Inflated-Beam Robot.** The total length and weight of the soft robot arm is 44 cm and 0.12 kg, respectively. The main body has a radius of 3.8 cm. As described by Naclerio and Hawkes [30], the beams forming the main body and the fPAM are constructed by forming tubes with bias-cut, woven fabric. The bias-cut orients the fabric fibers such that the tubes become shorter and wider when pressurized. This results in the shortening during pressurization. We use two pressure regulators (Festo VPPI) to control the pressure in the main body and fPAM, and each regulator also has a built-in pressure sensor. The main body is held at a constant pressure (0.4 kPag), and the fPAM is commanded to varying pressures (-0.7 to 20 kPag). The lower body pressure reduces opposition to fPAM actuation while having enough pressure to maintain its cylindrical shape. The fPAM command bounds were determined empirically with the following principles: (1) the minimum pressure must show the fPAM visibly deflated (2) the maximum pressure must cause maximum contraction of the fPAM (3) the range of pressure commands should be small to increase tracking performance of the pressure regulator. Our central computer sends pressure commands and receives pressure measurements via serial communication with a microcontroller (Teensy 3.6).

**Mobile Base.** We utilize a cart on a belt-driven linear actuator (Igus ZLW-1040B) that acts as a mobile base for our soft robot arm. The base of the soft robot arm is directly attached to the cart, which slides along the actuator rails within a 0.6 m range. We use the linear actuator's "Velocity Mode", which requires setting an acceleration value followed by sending velocity targets over time. Our central computer communicates with the linear actuator over Transmission Control Protocol (TCP).

**Motion Capture Sensing.** To simplify the hardware and sensing scheme for our robot, we use a motion capture system (OptiTrack with Flex 13 cameras) to measure the robot's current configuration. We place five sets of markers equally spaced along the soft robot arm, and a sixth set on the sliding cart. For each set of markers, the motion capture system provides the position and orientation in the global frame. We also use motion capture to measure task-specific observations $\mathbf{o}_{\text{task}}$.

**Simulation and Policy Learning Details.** We simulate the soft robot arm using Isaac Gym [31], a high-performance simulator that leverages GPU parallelization to simulate thousands of robots simultaneously. Using one NVIDIA RTX 3090 GPU, our simulation runs at 18,000 FPS (each frame is one action step with a control timestep of 33 ms) by running 4,096 environments in parallel. The simulation timestep and control frequency are two important parameters to determine. Our real, physical system runs at a 30 Hz control frequency (most of this time is spent communicating with the

sensors to measure the current state), so we run the control policy at 30 Hz in simulation accordingly. However, simulation often requires smaller timesteps to ensure numerical stability. We found that simulating the vine robot at 1200 Hz (0.833 ms timestep) is sufficiently stable.

For modeling the fPAM pressure, we found that the filtering parameter $\alpha$ was different for inflation ($a_p > p$) and deflation ($a_p < p$). Thus, we use $\alpha_{\text{inflate}} = 0.86$ and $\alpha_{\text{deflate}} = 0.81$ for modeling inflation and deflation, respectively. For modeling the cart dynamics, we use $k_{\dot{v}} = 0.3$, $k_v = 30$, and an action delay of 1 control timestep (33 ms). For domain randomization, we use $\sigma_{\text{obs}} = 0.001$, $\sigma_{\text{act}} = 0.001$, and $\epsilon_{\text{dyn}} = 0.001$.

We train all learned policies with a learning rate of 3e-4, a discount factor $\gamma$ of 0.99, and a PPO clipping interval $\epsilon_{\text{clip}}$ of 0.2. We also normalize the observations, values, and advantages, and we train the policy with 4 epochs per policy update. Using a horizon length of 16 (number of timesteps between updates for each robot, with all robots running in parallel), 4096 simulated robots, and a maximum of 500 update iterations, the approximate number of training timesteps is 32M steps ($16 \times 4096 \times 500$). Training takes about 40-80 minutes on an NVIDIA RTX 3090 GPU, which is substantially less time than Elastica's $\sim$11 hour RL training [2].

The policy is fed normalized observations (subtract the mean and divide by the standard deviation of each dimension). It outputs actions in $[-1.0, 1.0]$, which are then scaled to be in the appropriate range for each action dimension. We train the policy with the Proximal Policy Optimization (PPO) algorithm [32], using a highly-optimized GPU implementation called rl_games [33], which uses vectorized observations and actions for faster training.

## F  Domain Randomization Details

In our experiments, we use $\sigma_{\text{obs}} = 0.01$, $\sigma_{\text{act}} = 0.005$, and $\epsilon_{\text{dyn}} = 0.05$. Note that the observation noise and action noise are applied to the normalized observation and the unscaled action (actions in the range $[-1.0, 1.0]$). This ensures that the scale of the noise relative to the original value is consistent, so all components are affected similarly.

## G  Evaluating Failures

When evaluating our learned policy without fPAM smoothing, we noticed that the real fPAM actuations were not well-coordinated with the cart's motion. From there, we compared the commanded and actual fPAM pressures and found that the actual pressure did not immediately track the commanded pressure. When evaluating our learned policy without the linear actuator adjustments, we noticed that the real cart would not be able to change direction quickly enough to avoid the rail limits. We compared the sim and real cart velocity tracking profiles, which helped us model the cart controller more accurately. Finally, when evaluating our learned policy with ground truth velocities, comparing real and sim observations revealed that the velocities computed with first-order finite-differencing were noticeably different from the simulator's ground truth velocities.

## H  Trajectory Optimization Planning and Control Method

We use trajectory optimization to determine a reference trajectory (control trajectory $\mathbf{u}^{1:N-1}$ and resulting state trajectory $\mathbf{x}^{1:N}$) that brings the tip of the soft robot arm to the goal position within a fixed time horizon $N$. We define the state vector $\mathbf{x} := (y_{\text{cart}}, \dot{y}_{\text{cart}}, \theta_1, \dot{\theta}_1, \ldots, \theta_5, \dot{\theta}_5)$ and control input $\mathbf{u} := (F_{\text{cart}}, a_p)$ where $F_{\text{cart}}$ is the force applied to the cart and $a_p$ is the fPAM pressure. We solve the following optimization problem to compute the reference trajectories.

$$\min_{\mathbf{x}^{1:N}, \mathbf{u}^{1:N-1}} \quad \sum_{k=1}^{N} \left\| \mathbf{x}^k - \bar{\mathbf{x}}^k \right\|_{Q^k}^2 + \sum_{k=1}^{N-1} \left\| \mathbf{u}^k \right\|_{R^k}^2$$

$$
\begin{aligned}
\text{s.t.} \quad & \mathbf{x}^{k+1} = f(\mathbf{x}^k, \mathbf{u}^k), && k = 1, \ldots, N-1, \\
& \mathbf{u}_{\min} \le \mathbf{u}^k \le \mathbf{u}_{\max}, && k = 1, \ldots, N-1, \\
& y_{\text{cart-min}} \le y_{\text{cart}}^k \le y_{\text{cart-max}}, && k = 1, \ldots, N-1, \\
& \dot{y}_{\text{cart-min}} \le \dot{y}_{\text{cart}}^k \le \dot{y}_{\text{cart-max}}, && k = 1, \ldots, N-1, \\
& g(x^N) = 0
\end{aligned}
\tag{4}
$$

where $y_{\text{cart}}^k$ and $\dot{y}_{\text{cart}}^k$ are the first and second elements of $\mathbf{x}^k$ respectively. The objective function is a quadratic cost on deviation from a nominal state trajectory $\bar{\mathbf{x}}^{1:N}$ with weight matrices $Q^{1:N}$ and a quadratic cost on control effort with weight matrices $R^{1:N-1}$. The first constraint is for dynamic feasibility, the second is for control limits, the third is for cart position and velocity limits, and the fourth is for the tip position to reach the goal position at the final timestep. We set each $\bar{\mathbf{x}}^k$ of the nominal trajectory to be a pose that bends to the left (with zero velocity) since all goal positions require bending to the left. Because we use a multi-link rigid body approximation as our dynamic model, we have a broader array of options for our simulator and optimizer choice. For this work, we chose Dojo [34] for its numerical stability (since we have stiff equations of motion and would like to take larger time steps), and we used its associated trajectory optimization package which implements iterative Linear Quadratic Regulator with Augmented Lagrangian methods.

The numerical values for $Q^k$ and $R^k$ for $k = 1, \ldots, N-1$ are:

$$diag(Q^k) = (10, 1, 10, 1, 10, 1, 10, 1, 10, 1, 10, 1)$$
$$diag(R^k) = (1, 1)$$

The numerical values for $Q^N$ are:

$$diag(Q^N) = (100, 10, 100, 10, 100, 10, 100, 10, 100, 10, 100, 10)$$

Dojo is a powerful simulator in that it is able to simulate robots with stiff dynamics while taking relatively large time steps, while remaining numerically stable (in contrast to IsaacGym, which needed to be simulated at 1200 Hz to avoid numerical instability). This is critical for reducing the number of timesteps in the optimal trajectory (and in turn the number of variables in the optimization problem). However, it has yet to be optimized for runtime performance. Solving the trajectory optimization problem took between 1-13 minutes, depending on the target position. We expect the solve time to decrease as the simulator is further optimized for speed. Re-planning in real-time, (e.g. with model predictive control), was not possible due to the optimization solve time, so instead we used a simple but fast tracking controller that adds a feedback term to the reference action trajectory based on deviation from the reference tip trajectory:

$$a_{\text{cart-vel}}^k = a_{\text{cart-vel ref}}^k + k_y(y_{\text{tip ref}}^k - y_{\text{tip}}^k) \tag{5a}$$

$$a_p^k = a_{p\,\text{ref}}^k + k_z(z_{\text{tip ref}}^k - z_{\text{tip}}^k). \tag{5b}$$

Reference action trajectories $a_{\text{cart-vel ref}}^{1:N-1}$ as well as $a_{p\,\text{ref}}^{1:N-1}$ and reference tip trajectories $y_{\text{tip ref}}^k$ and $z_{\text{tip ref}}^k$ are extracted from the optimal solution for $\mathbf{x}^{1:N-1}$ and $\mathbf{u}^{1:N-1}$. The current tip position $(y_{tip}^k, z_{tip}^k)$ is measured with motion capture, and $k_y$ and $k_z$ are controller gains. The adjusted actions are clamped to be within the action limits for the cart and fPAM and then sent to the physical hardware. During trajectory optimization, we use conservative constraints on cart velocity and fPAM pressure so that the tracking controller has margin to exceed the reference control before needing to be clamped within actuator limits. We empirically chose gains that improved performance for one of the more difficult target positions, and then used this for all other experiment runs. The gains used were $k_y = 0.1$ and $k_z = 5.0$.

As with our policy trained with RL, we require careful but simple strategies to achieve sim-to-real transfer. First, the trajectory constraints allow us to respect state and control limits (i.e. cart position, velocity, and acceleration as well as fPAM pressure). We note that we indirectly enforce constraints on cart acceleration by constraining $F_{cart}$. Second, the use of a tracking controller allows us to overcome minor model errors, similar to how domain randomization during policy training produces robustness to model errors. Finally, we did not model actuator dynamics when solving the reference trajectory for simplicity, but this caused a delay between the reference and actual tip trajectory that could not be overcome by the tracking controller alone. We address this by extending the state and control reference trajectories for a few extra timesteps and filling the new elements with $x^N$ and $u^{N-1}$, respectively. We hypothesize that this adjustment handles the sim-to-real gap introduced by actuator latency and response times.

We visualize an example state and control trajectory in Figs. 11-14 (same trajectory as the one shown in Fig. 8). The tip position plots show the extension of the nominal trajectory by about 0.1 s. The plot of tip y-position shows how the additional time is necessary for the robot to reach the final position. In addition, our trajectory optimization formulation includes a quadratic cost on control effort, resulting in relatively smooth control trajectories as seen in Figs. 13-14. This aids in making both cart velocity and fPAM pressure reference trajectories easier to track. We also note that the actual tip position trajectories closely match the nominal trajectories, except that there is a slight delay that shifts the actual trajectories slightly to the right. We also see that the actual fPAM and cart velocity trajectories closely match their commanded trajectories but with a slight shift to the right due to actuator latency. We find that this planning and control method still performs well, despite this delay from actuator latency.

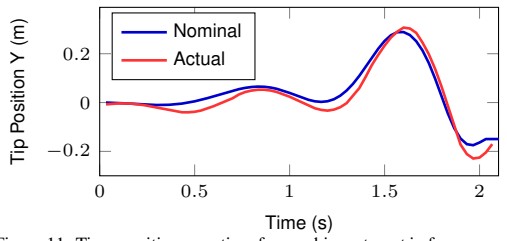

Figure 11: Tip y-position over time for reaching a target in free space. The nominal trajectory (blue) was computed with trajectory optimization. This motion plan is deployed on the physical hardware with a tracking controller and the actual trajectory is visualized in red.

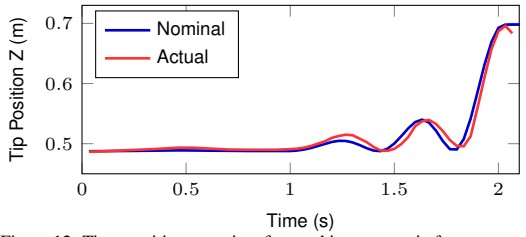

Figure 12: Tip z-position over time for reaching a target in free space. The nominal trajectory (blue) was computed with trajectory optimization. This motion plan is deployed on the physical hardware with a tracking controller and the actual trajectory is visualized in red.

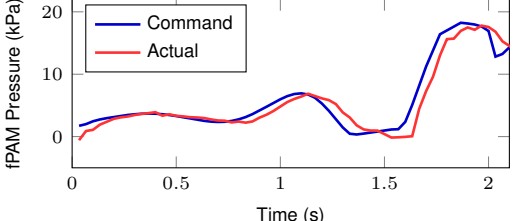

Figure 13: FPAM pressure over time for reaching a target in free space. The blue trajectory shows the actuator commands when tracking the reference trajectory on hardware. The red trajectory shows the actual fPAM pressure.

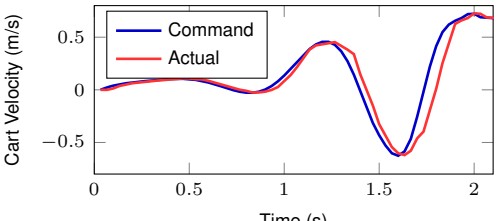

Figure 14: Cart velocity over time for reaching a target in free space. The blue trajectory shows the actuator commands when tracking the reference trajectory on hardware. The red trajectory shows the actual cart velocity.

## I  Comparison to PID Control for Free Space Target Reaching Task

We compare our learned policy to a proportional-integral-derivative (PID) controller on the free space target reaching task. This serves as a baseline controller that does not reason about leveraging swinging, and we show that it is largely unsuccessful in this task.

We use two separate PID controllers to achieve tip-position control. We use cart actuation to drive the y-coordinate error towards zero, and we use fPAM actuation to drive the z-coordinate error

towards zero. We acknowledge that the fPAM actuation also affects the tip y-coordinate, but find that the y-coordinate PID controller is able to account for this disturbance. Below are the equations used to compute the PID control commands (actions) for cart velocity and fPAM pressure:

$$a_{\text{cart-vel}} = -K_{p,y}e_y - K_{d,y}\dot{e_y} - K_{i,y}\int e_y dt$$

$$a_p = -K_{p,z}e_z - K_{d,z}\dot{e_z} - K_{i,z}\int e_z dt$$

where $e_y = y_{\text{tip}} - y_{\text{target}}$ and $e_z = z_{\text{tip}} - z_{\text{target}}$ are the y-error and z-error of the tip position, respectively. $K_*$ are the PID controller gains. We took a manual approach to tuning our PID gains that is similar to the Ziegler-Nichols method. Our final gains were $K_{p,y} = 1$ and $K_{p,z} = 20$. We found that derivative and integral gains had little effect on overall performance, and that increasing these gains led to instability, so we ultimately set these to zero.

We run this PID controller for the same 54 target positions discussed in Sec. 4, and the robot achieves a success rate of 17% across the 54 target positions. With this control method, the robot is not able to reach any of the target positions with a z-coordinate of 0.6 m or greater; the maximum tip z-coordinate reached across all PID control experiments was 0.56 m. The PID controller limits the robot's workspace because it aims to greedily reduce tip position error and does not incorporate any reasoning about swinging, significantly reducing its ability to reach higher target positions. We illus-

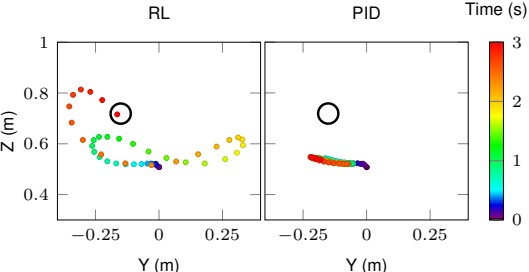

Figure 15: Comparison of learned policy (RL) vs. PID control (PID) for reaching a target tip position in free space. The black circle is centered at the target position with a radius of 4 cm. The RL control policy is able to perform high-speed swinging behavior to reach the target position. The PID control is unable to track the target position because it greedily approaches the target position directly.

trate this in Fig. 15. Using our learned policy, the robot is able to reach the target tip position by building up momentum over multiple swings to increase its tip height. In contrast, using the PID controller, the robot is unable to reach the target tip position, as it simply moves directly towards the target and fails to bring the tip high enough. This demonstrates that the free space target reaching task is not possible without well-timed actuation that leverages swinging motion.

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
