# OpenReview forum: "Reinforcement Learning Enables Real-Time Planning and Control of Agile Maneuvers for Soft Robot Arms"
_robot-learning.org/CoRL/2023/Conference — CoRL 2023 Poster_

### Official Review · Reviewer_y2wh · 2023-07-10

**Confidence:** 3
**Originality:** Fair
**Technical Quality:** Good
**Clarity Of Presentation:** Very Good
**Impact:** 3

**Recommendation:**

Weak Accept: I recommend accepting the paper, but will not argue for my recommendation if the majority of other reviewers have a different opinion.

**Review:**

# Strength
- Overall, this paper has good clarity, is well-written, and is easy to follow. The figures are beautiful. The demo video is impressive and helps the readers' understanding.
- Great engineering effort to make the sim-to-real transfer of soft robots tractable by considering the soft robot, linear actuator, and pressure regulator dynamics. Demonstrating dynamic control for soft robots using sim-to-real approaches should be highly respected.
- Good quality: comparisons and ablation studies are explored in detail.

# Weakness
- My primary concern is weak originality and the lack of technical contribution to learning parts. It seems just to apply domain randomization techniques, which have been widely used in previous studies. Considering actuator dynamics for sim-to-real may not be a substantial contribution. For example, [r1] considers actuator dynamics for sim-to-real transfer. It would be natural to consider the actuator dynamics even in soft robots.
   - [r1] Miki, T., Lee, J., Hwangbo, J., Wellhausen, L., Koltun, V., & Hutter, M. (2022). Learning robust perceptive locomotion for quadrupedal robots in the wild. Science Robotics, 7(62), eabk2822.
- This study has weak significance and relevance. Although it poses an interesting problem, dynamic soft robot manipulation, it may not address the problem of robot learning.
Besides, it solves this problem with the existing approaches mentioned above. This study applies a learning technic but does not improve it. CoRL audiences may not be interested because this study mainly introduces actuator modeling.
- This study needs to add more citations about sim-to-real transfer for soft robots and discuss why the existing sim-to-real approach is difficult. For example, the following literature [r2-4] may be close to this study.
   - [r2] Dubied, M., Michelis, M. Y., Spielberg, A., & Katzschmann, R. K. (2022). Sim-to-real for soft robots using differentiable fem: Recipes for meshing, damping, and actuation. IEEE Robotics and Automation Letters, 7(2), 5015-5022.
  -  [r3] Hagiwara, K., Yamamoto, K., Shibata, Y., Komagata, M., & Nakamura, Y. (2022). On high stiffness of soft robots for compatibility of deformation and function. Advanced Robotics, 36(19), 995-1010.
  -  [r4] Li, Yingqi, Xiaomei Wang, and Ka-Wai Kwok. "Towards Adaptive Continuous Control of Soft Robotic Manipulator using Reinforcement Learning." 2022 IEEE/RSJ International Conference on Intelligent Robots and Systems (IROS). IEEE, 2022.
- I respect the engineering efforts. Also, the combination of DR and actuator dynamics showed better performance.  However, it also implies that this study provides many assumptions to achieve the sim-to-real transfer. The ablation studies revealed that actuator and sensor models were more impactful than domain randomization.



**Quality Of The Limitations Section:**

Limitations are addressed clearly

**Questions For Rebuttal:**

- Could you discuss the novelty of the learning part by referring to the other citations about sim-to-real transfer for soft robots?
- The filter parameter of the regulator and latency of the linear actuator is designed experimentally. Thanks to them, it is good to see the simulator behavior is close to the real one. What happens if these parameters are randomized during training? Will the policy become more robust?
- It will be better to show what parameters were randomized in not the supplemental material but the manuscript.
- Did this study consider the filtering of the regulator and latency of the linear actuator for trajectory optimization? If not, testing trajectory optimization with them will be a more fair comparison.
- Could the proposed approach perform trajectory tracking tasks?

**Robotics Focus:**

Sufficient demonstration on hardware

**Summary Of Paper:**

This paper presents a sim-to-real transfer approach for dynamic control of soft robot arms. It prepares a linear dynamics model for fabricated Pneumatic Artificial Muscles (fPAM) where the dynamics parameters are identified with the 30s of the real-world data. Then, domain randomization is applied to the dynamics parameters of the linear dynamics model during RL training.  The real robot experiments showed that the proposed method achieved higher success rates than the PID and trajectory optimization-based approaches. The ablation studies revealed that the combination of all proposed components improved the performance.

**Summary Of Recommendation:**

I respect the engineering efforts for the sim-to-real transfer; however, this study would lack the technical contribution for machine learning parts. For this reason, I would like to recommend a weak rejection of the current manuscript.

After the rebuttal comment: I raise my recommendation from weak reject to weak accept. Although the technical contribution of the learning part is weak, the simple simulator enabling soft robot sim-to-real is good news and benefits the robot learning community.

---

### Official Review · Reviewer_UnPS · 2023-07-17

**Confidence:** 4
**Originality:** Good
**Technical Quality:** Good
**Clarity Of Presentation:** Very Good
**Impact:** 3

**Recommendation:**

Weak Accept: I recommend accepting the paper, but will not argue for my recommendation if the majority of other reviewers have a different opinion.

**Review:**

The problem of using RL with soft robotics is important for robotics. Soft robots are desirable for many reasons but they are hard to control. The authors introduce these benefits and challenges clearly.

The proposed method is clearly described and tested on the hardware. The results are promising and ablation studies are also very useful to identify what is crucial for the policy.  My only concern about the paper is the simplicity of the experimental setup. If the authors used a more complex robot with a different form factor, the paper could be much more impressive. As an example,
I would be very interested in to see the results if the problem studied was very sensitive to the force applied to the tip (which would cause deformation of the robot due to external forces). In any case, the results are well analyzed and the authors show that RL can be used on a soft robot without an accurate simulation model. The paper is definitely has some nice contributions and opens interesting future resarch directions.

The plots are not possible to understand when printed in Black and White.

**Quality Of The Limitations Section:**

Limitations are addressed clearly

**Questions For Rebuttal:**

Can do authors show some behaviors on more complex problems (i.e. flipping a piece of paper, dragging a cloth, etc) or form factors (quadruped with soft legs)?

**Robotics Focus:**

Highly relevant to robotics but no hardware experiments

**Summary Of Paper:**

The paper aims training control policies for soft robots using reinforcement learning and simulation. The soft robots are very hard to develop a high fidelity simulator, so RL is not easy to use with. The authors show that they can solve the problem for a single joint simple soft robot arm attached to a linear actuator. The authors model the arm with N-joint arm simulation, use domain randomization and careful tuning of simulation parameters such as latency. The results show that the trained policy can reach to different tip points, and the author show results on hardware as well. The ablations also show that different components are crucial for success.

**Summary Of Recommendation:**

I recommend the acceptence of the paper because it shows interesting results, but the evaluation could be much better if the authors used more complex problems or form factors.

---

> ### Author Response · Authors · 2023-08-11
> **Response to Reviewer UnPS**
>
> # Response to Reviewer UnPS:
>
> We express our gratitude to the reviewer for their valuable feedback, which has helped us improve our work. Below we address each of the comments and questions:
>
> __Comment/Question: “My only concern about the paper is the simplicity of the experimental setup. If the authors used a more complex robot with a different form factor, the paper could be much more impressive. As an example, I would be very interested in to see the results if the problem studied was very sensitive to the force applied to the tip (which would cause deformation of the robot due to external forces) ... Can do authors show some behaviors on more complex problems (i.e. flipping a piece of paper, dragging a cloth, etc) or form factors (quadruped with soft legs)?”__
>
> __Response:__ While it is true that our soft robot arm platform features fewer actuators than many other soft robots, our results suggest that RL can be applied in the real world to many soft robots without requiring a high-fidelity simulation model. We believe our contribution is an important first step for soft robot arm control and opens many doors for future work. We agree with the reviewer that there are compelling research questions in scenarios when forces are applied to the tip of the arm. For example, we are actively working on applying our framework to a 3D context, where a soft robot arm mounted on a drone can perform aerial manipulation tasks requiring interaction with the environment.
>
> __Comment/Question: “The plots are not possible to understand when printed in Black and White.”__
>
> __Response:__ We apologize for this, and we are grateful to the reviewer for bringing this to our attention. If given the opportunity, our camera-ready version will feature plots with varying line styles to resolve this.

---

> > ### Comment · Reviewer_UnPS · 2023-08-15
> > **Response to Response**
> >
> > Thanks to authors for their response. I agree that the paper opens up many interesting research directions and the experiments support this idea and they are in the right direction. It is a very interesting and well written paper. But I still think that the evaluations could be stronger / more complex for me to switch my recommendation to "strong accept, so I'll leave my recommendation as it is.

---

### Official Review · Reviewer_sYng · 2023-07-20

**Confidence:** 4
**Originality:** Very Good
**Technical Quality:** Excellent
**Clarity Of Presentation:** Excellent
**Impact:** 3

**Recommendation:**

Strong Accept: I recommend accepting the paper and will argue for my recommendation even if other reviewers hold a different opinion.

**Review:**

I am impressed by this paper. The idea is formulated in a clear way, the results are convincing and the ablations show the effects of almost all components.
I enjoyed watching the video, which presented the experiments very clearly.
While the "soft" control aspect of the arm is quite limited, due to the low-dimensional nature of the setup, and there remain minor issues with framing and one or two missing explanations in the text, I think this paper is a very well executed step towards better control methods for soft actuators.

Finally, the limitations section should put stronger emphasis on the fact that a low-dimensional system was employed. It is very likely that more complex soft robots will require more components that specifically deal with the sim2real gap in the soft structures.

Detailed comments:
Figure 2 is not mentioned in the text. I think it would also further understanding if the policy architecture was explained in a more detailed way in text somewhere. By the time I read about the ablations, I had almost forgotten about the recurrent architecture of the policy. These choices should be stated more clearly, similar to the other components.

Related Work: Potentially add some explanations on how soft robot control is to be distinguished from soft actuation. Robots like the one in [A] also use soft/muscle-like actuators, but are otherwise rigid. These robots have already achieved highly dynamic behaviors, so it would be good to discuss how the employed soft robot is different.

Line 169: Add citations about IIR use of pressure modelin for soft robots

Line 178: We design a a cart controller [...]
This paragraph is written in a confusing way. You mention that a constant force cannot be applied to the cart. Is this meant to apply to simulation or to the real system?. In simulation one can set any acceleration directly. Perhaps it would be better to argue that the real car cannot reach any velocity instantly and therefore the controller in simulation has to be adapted in order to match the real system better. One way of doing that is the velocity tracking/acceleration tracking controller.

Figure 6: Maybe also plot the simulated cart velocity when no delay is added. That would make the magnitude of the effect clearer.

Section 4.2: Consider marking the goals that were used in the ablations in Figure 7.

Line 275: What exactly is "domain randomization on policy actions"? I cannot find the exact explanation of this.

Finally, consider adding a disclaimer to the video: "Turn sound on", I missed it the first time.



[A] Learning to Play Table Tennis From Scratch using Muscular Robots. Büchler et al.

**Quality Of The Limitations Section:**

Additional details required

**Questions For Rebuttal:**

Q1: Section 3.3: It is interesting that so few of the proposed components deal directly with the soft part of the robot, most seem to increase simulation accuracy for the cart actuation. Was the dynamics estimation for the soft part already good enough?

Q2: Why did the planning stage take 4.7 hours? I understand that real-time planning is quite difficult to achieve with TO, but this planning time seems excessive. Could you explain further?



**Robotics Focus:**

Sufficient demonstration on hardware

**Summary Of Paper:**

This paper aims to learn a reinforcement learning policy for a mixed actuator soft robot. The policy is trained in simulation and several adjustments to the simulation dynamics in combination with a system identification approach and a recurrent policy architecture are proposed in order to increase the performance of the controllers after transfer to the real system. The final policy is able to achieve many goal positions for which a combination of linear cart and soft actuator control is required. Several ablations are performed which demonstrate that the proposed extensions to the simulation engine are pertinent for performance. A PID controller baseline does not perform well, while a trajectory-optimization baseline is used to establish an upper performance bound---at strongly increased computational cost.

**Summary Of Recommendation:**

This paper is clearly written and well executed. The problem of sim2real transfer is important and while the study only addresses a simple soft system, the proposed results are well motivated and convincing. Good paper.

---

### Decision · Program_Chairs · 2023-08-30

**Decision:**

Accept (Poster)

**Comment:**

The paper presents a reinforcement-learning- and sim-to-real-based approach to control a soft robot.

Reviewers appreciated the clear formulation, strong motivation, tests of physical hardware, and video. Achieving sim-to-real transfer for soft robots with a simple simulator is an excellent contribution. Reviewers also praised the papers writing quality, the video, and the thoroughness of ablation studies.

Reviewers raised concerns that the simplicity of the robot in experiments may suggest limitations on the proposed approach that should be more clearly stated in the limitations. A reviewer also provided a detailed related work that the authors should consider including.

Reviewers also appreciated the author's rebuttal, raising at least one reviewer's rating. As a result, all reviewers agree that the paper is above-bar and should be accepted.